# Assessment of the Nutritional Status and Quality of Life in Chronic Kidney Disease and Kidney Transplant Patients: A Comparative Analysis

**DOI:** 10.3390/nu14224814

**Published:** 2022-11-14

**Authors:** Weronika Pawlaczyk, Lukasz Rogowski, Joanna Kowalska, Małgorzata Stefańska, Tomasz Gołębiowski, Oktawia Mazanowska, Claire Gerall, Magdalena Krajewska, Mariusz Kusztal, Wioletta Dziubek

**Affiliations:** 1Lower Silesian Oncology Center, Department of Physiotherapy, 53-413 Wroclaw, Poland; 2Faculty of Health and Physical Culture Sciences, The Witelon Collegium State University, 59-220 Legnica, Poland; 3Faculty of Physiotherapy, Wroclaw University of Health and Sport Sciences, 51-612 Wroclaw, Poland; 4Department of Nephrology and Transplantation Medicine, Faculty of Medicine, Wroclaw Medical University, 50-367 Wroclaw, Poland; 5Division of Pediatric Surgery, Department of Surgery, Columbia University Vagelos College of Physicians and Surgeons, NewYork-Presbyterian Morgan Stanley Children’s Hospital, New York, NY 10032, USA

**Keywords:** hemodialysis, kidney transplant, MNA, malnutrition, quality of life, KDQoL

## Abstract

Background: Chronic kidney disease (CKD) can significantly influence a patient’s nutritional status, leading to malnutrition. Malnutrition is associated with an increase in morbidity and hospital admissions, as well as a decrease in functional status. All these factors impact emotional, physical, and psychosocial health, leading to a lower quality of life (QOL). The aim of the study was to assess the nutritional status and QOL in patients with CKD compared to patients after kidney transplantation and determine what factors influence nutritional status and QOL in this patient population. Methods: The study included 167 patients: 39 pre-dialysis patients—group 1; 65 dialysis patients—group 2; 63 kidney transplant patients—group 3. Patients completed the Kidney Disease Quality of Life questionnaire (KDQoL) and the Mini Nutritional Assessment questionnaire (MNA). Results: A comparative analysis of the QOL of patients in the three study groups showed no statistically significant differences in the overall KDQoL scores. Factors that affected quality of life included the designated group, determined by disease status, MNA score, patient age, and WHR. Nearly 1/3 of patients from groups 2 and 3 were at risk of malnutrition. Conclusions: A systematic assessment of nutritional status and monitoring of QOL should be integrated into the standard management guidelines for CKD patients.

## 1. Introduction

Chronic kidney disease (CKD) is associated with a high risk of morbidity and mortality, and presents a significant public health concern, with increasing incidence and prevalence. It is estimated that 600 million people worldwide suffer from CKD, with approximately 4.2 million in Poland [1]. Like any chronic disease, in addition to physical manifestations, CKD can affect an individual’s social, professional and family life. Previous studies have shown that the quality of life (QOL) of patients with CKD is reduced compared to the healthy population [1]. With progression to end-stage renal disease (ESRD), patients become dependent on dialysis, a time-consuming therapy that greatly affects lifestyle. Many of these patients become preoccupied with the hope of receiving a kidney transplant, allowing for them to function normally and improving their QOL.

In addition to a decreased QOL, malnutrition is common in hemodialysis (HD) patients, with prevalence ranging from 18 to 75%, and continues after renal transplant [2,3]. Hemodialysis treatments carry a risk of malnutrition due to the catabolic effects of renal replacement therapy, dietary restrictions, loss of nutrients through the dialysis membrane, inflammation, and metabolic acidosis that can lead to protein energy-wasting (PEW) [2,4,5]. PEW, a term introduced by the International Society of Renal Nutrition and Metabolism (ISRNM), refers to many of the nutritional and catabolic changes that occur in CKD patients, which are associated with increased morbidity and mortality [2,6]. Poor nutrition also affects QOL, as it is associated with increased morbidity, decreased functional capacity and an increased number and duration of hospitalizations in this patient population. Several studies have shown that malnourished patients have a worse QOL, making the early diagnosis and treatment of malnutrition essential [7,8,9,10].

As a result, diet is a fundamental aspect in the management of CKD patients, with dietary regimens being the most restrictive of all chronic diseases [11,12]. Dietary modifications aid in the prevention and treatment of PEW, electrolyte imbalances, and bone and mineral abnormalities. Modifications initiated in the early stages of CKD may slow the progression of disease while, in later stages, this may delay the need for renal replacement therapy [11,12,13]. In patients who received a kidney transplant, nutrition often improves, which is shown to be associated with improved QOL [14]. Malnutrition in this subset of patients increases the risk of infection, delayed wound-healing, and muscle weakness [15].

The aim of the study was to assess the nutritional status and quality of life in patients with CKD compared to patients after kidney transplantation. The secondary aim was to determine what factors influence nutritional status and QOL in this patient population.

## 2. Materials and Methods

### 2.1. Participants

This observational study was conducted at the University Clinical Hospital in Wroclaw, Poland between 2018 and 2019. Patients with stage III-IV CKD who were not dialysis-dependent, patients receiving HD for at least 6 months, and patients at least 6 months post-kidney transplantation were included in this study.

Participant Inclusion criteria:-Consent to participate in the study;-Age 45–75 years;-Have a diagnosis of chronic kidney disease;-Hemodialysis treatment for at least 6 months;-Kidney transplant (at least 6 months after the transplant).Participant Exclusion criteria:-Transplant rejection within the last six months;-Unwilling to participate in the study.Designated Patient Groups:Group 1—39 pre-dialysis patients with stage III or IV CKD (mean age 58.56 ± 8.04);Group 2—65 dialysis patients with stage V CKD (mean age 60.49 ± 7.57; mean dialysis time 110 months);Group 3—63 kidney transplant patients (mean age 62.06 ± 6.98; mean 118 months after transplantation).The study was conducted in accordance with the Helsinki Declaration and under the ethical and legal supervision of the Bioethics Committee of the Wroclaw University of Health and Sport Sciences, Poland (reference no 26/2017). All participants were informed of the purpose and methods of the study and the ability to withdraw at any time. Patient consent was obtained prior to enrolling in the study.

### 2.2. Measurements Tools

#### 2.2.1. The Kidney Disease Quality of Life Short Form (KDQoL-SF^TM^)

This questionnaire is used to assess the QOL of patients with kidney disease, including those undergoing renal replacement therapy. It contains twenty-four questions covering a wide range of domains of human functioning in daily, occupational and social life. It is a self-report measure, resulting from a subjective perception of one’s health, functionality, and their QOL as directly related to having kidney disease. Hemodialysis patients additionally rate their satisfaction with the care they receive for HD procedures [16]. Each answer in the questionnaire is assigned a specific number of points. The scoring of answers in the KDQoL-SF^TM^ questionnaire is carried out using a scale from 0 to 100 points. The higher the value, the higher the QOL, and the lower the value, the worse the QOL [17].

The kidney disease summary component (KDSC) includes symptom/problem list, effects of kidney disease, burden of kidney disease, work status, cognitive function, quality of social interaction, sexual function, sleep, social support, dialysis staff encouragement, and patient satisfaction. The physical component summary (PCS) and the mental component summary (MCS) scores from the SF-36 questionnaire include physical functioning, role-physical, pain, general health, emotional well-being, role-emotional, social function, and energy/fatigue.

#### 2.2.2. Mini Nutritional Assessment Questionnaire (MNA)

The MNA is a validated screening tool, which was originally developed to assess the nutritional status of elderly patients. This scale consists of an abbreviated six-item screening version (MNA-SF) and a full eighteen-item scale (MNA-LF).

The full form consists of 18 questions to assess 4 different aspects: anthropometric assessment (BMI, arm and calf circumferences, weight loss), general assessment (lifestyle, medications taken, mobility, presence of symptoms of depression or dementia), nutritional assessment (number of meals, amount of protein intake, amount of fluid intake, nutritional autonomy) and subjective patient assessment (self-assessment of nutritional status, self-assessment of health compared to peers).

The maximum total score is 30 points, with established cutoff values determining the risk of malnutrition, as seen below:24–30 points—normal nutritional status;17–23.5 points—risk of malnutrition;<17 points—malnutrition [18,19,20].

### 2.3. Statistical Analysis

The Shapiro–Wilk test was used to check the distribution of quantitative variables. Descriptive statistics were calculated. Due to the non-normality of the distribution of some quantitative variables and the qualitative character of some variables, the mean and median were used as a measure of central tendency, and standard deviation (SD) and interquartile range (IQR) as a measure of dispersion. Significance between study groups was determined by one-way ANOVA, Kruskal–Wallis test or Chi-square test. The significance level was (*p* < 0.05). Variables with significant differences between groups underwent post hoc test (Tukey or Dunn Bonferroni–Holm). Logistic regression and multivariate regression were performed (keeping the distribution of residual variables normal) to indicate which factors have the greatest impact on nutritional status and QOL. Calculations were performed using Statistica 13.3 and PQ Stat 1.8.2. All calculated multivariate regression models were characterized by a high fit (adjusted R^2^ about 50%) and statistical significance (*p* < 0.05). The normality of the distribution of residuals was confirmed in all analyses.

## 3. Results

A total of 167 patients were included in this study, with a mean age 60.6 (±7.5) years. Patients were separated into three groups, as noted in the methods. Patient demographics did not significantly differ between the groups, as detailed in Table 1.

MCS scores significantly differed between patients in group 1 and group 3, with group 1 having significantly lower scores, while those in group 3 scored the highest (*p* < 0.05). KDCS scores significantly differed between groups 1 and 2 and groups 2 and 3 (*p* < 0.05), with the highest scores being observed in group 2. There was no statistically significant difference in PCS score between groups. However, it was observed that patients in group 2 scored, on average, 7 points lower than those in group 1 or 3 (Table 2).

Almost 1/3 of patients in group 2 and 3 had an MNA score indicating risk of malnutrition (MNA < 24). A summary analysis of MNA score showed the highest average score in group 1. Statistically significant differences were found between groups 1 and 2 and 1 and 3 (*p* < 005), as seen in Table 2. There were no significant differences between groups in terms of BMI, waist and hip circumference and WHR values (Table 1 and Table 3).

Multivariate regression models were utilized to see which factors influence the different components of the KDQoL-SF^TM^ questionnaire (Table 4). PCS was shown to be significantly influenced by age, WHR, MNA and MCS values. Patients younger than 65 years of age, with a lower BMI, and higher WHR, MNA and MCS scores, obtained higher PCS values (Table 5).

MSC was significantly influenced by assigned group (a surrogate for severity of disease), age, and PSC and KDCS levels. MCS scores were highest in group 1 patients, aged over 65 years, with higher PSC and KDCS scores (Table 6).

KDCS was shown to be significantly associated with age, WHR and MSC value. Higher KDCS scores were observed in patients with age less than 65 years, higher WHR and higher MCS scores (Table 7).

## 4. Discussion

A comparative analysis of the QOL of patients in the three study groups showed no statistically significant differences in the overall KDQoL scores. However, in regression analysis, group designation significantly affected KDQoL scores, with group 3 (kidney transplant patients) scoring the highest on average and group 2 (dialysis patients) scoring lowest. The direct correlation between renal function and perceived QOL is well documented in the current literature. It is observed that, whereas the renal function decreases, patients have a lower assessment of their quality of life [21,22,23], while, after kidney transplantation, KDQoL scores are higher [24]. The lack of significance within our data was likely influenced by an older post-transplant population compared to similar studies (mean age of 62.06 ± 6.98 years in our post-transplant patients compared to 45.3–55.3 years in the similar literature) [18,23]. With age, the influence of factors not directly attributable to the disease increases, which may decrease QOL scores [25]. Another explanation may be the presence of a large standard deviation in the scores of individual groups. Comparative analysis of the QOL of patients in the three study groups showed statistically significant differences in MCS and KDCS. Patients in group 3 (kidney transplant) reported the highest scores in MCS. This is consistent with previous reports by many authors [24,26,27]. Avramovic and Stefanovic report higher scores on the SF-36 among pre-dialysis CKD patients when compared to hemodialysis patients, with post-kidney transplant patients reporting the highest QOL [28]. Ryu et al. echoes these findings, highlighting a better QOL and improved survival in patients who underwent successful kidney transplantation when compared to dialysis patients. Notably, a lower QOL was independently associated with post-transplant morbidity and mortality [24]. In a study by Iqbal et al., transplant recipients had higher QOL scores, with similar physical and social function scores to healthy controls patients [29]. The results of Sarhan et al. also indicated that kidney transplant recipients have a better QOL than hemodialysis patients in some components of the SF-36, including social functioning and emotional role, but hemodialysis patients had better QOL in terms of physical functioning and physical role [30].

The physical status is the most affected component of QOL [31]. In our study, the group of dialysis patients had also a better QOL in PCS physical components (PCS). This group of patients had better KDCS results. Since this group has the best QoL in the area of physical functioning and the fewest associated limitations, it would seem that the time needed to treat and fight the disease and the frustration associated with the disease is less than that in the other groups. Nevertheless, renal disease is already associated with many limitations in patients’ functioning at home and in the community in the conservative period, which promotes the development of depression, anxiety and sleep disorders [32,33,34]. This is an emotionally difficult stage for a patient with a diagnosis of CKD, which, in turn, is explained by the lowest MCS scores.

Diet plays an important role in improving the quality of life of dialysis patients. In the case of renal replacement therapy, adherence to an appropriate diet has a major impact on the proper course of dialysis, well-being, the results of certain laboratory tests and the nutritional status of patients [35,36]. Chronic malnutrition is a common complication in CKD patients, affecting up to 80% of patients, depending on the study population (CKD stage) and assessment procedure [36,37,38,39]. Malnutrition may be the result of systemic inflammation, consequences of hemodialysis, or the insufficient consumption of substrates in the diet, either in the process of CKD prevention or as a result of several other factors (eating disorders, impaired taste sensation, mental disorders). CKD patients are advised to follow specific dietary restrictions, including a limitation of animal products (meat, dairy) due to their high protein and phosphorus content and reduced consumption of vegetables that are high in water and phosphorus [38,40]. These factors can lead to nutritional deficiencies, which may negatively impact the QOL and physical fitness of CKD patients [36,37,38,39]. Malnutrition and decreased QOL and physical fitness are significant risk factors for adverse outcomes and mortality in this population [35,41]. A low quality of life can also have a negative impact on compliance to therapeutic interventions [42].

The total result of the MNA questionnaire was significantly different between the pre-dialysis group and the other groups, with no differences for anthropometric indicators, i.e., body weight, BMI, or WHR. This is probably related to the nutritional habits of patients, as MNA questionnaires ask about the amount and type of meals and products consumed. Hemodialysis and post-transplant patients scored lower, most likely due a stricter adherence to dietary guidelines [20,38]. Our research aimed to determine the risk of malnutrition and the relationship between malnutrition and QOL in pre-dialysis, dialysis, and post-kidney transplant patients. Overall, the risk of malnutrition, assessed with the MNA questionnaire, was evident in all groups, and clearly higher for the HD group and kidney transplant patients. Almost 1/3 of HD and post-transplant patients had MNA scores indicating risk of malnutrition (MNA < 24). In addition to increased morbidity and mortality, malnutrition also results in a decreased QOL. Current management guidelines do not routinely include a systematic assessment of nutritional status and monitoring of QOL; however, patients would benefit from both prevention and the early detection and management of nutritional deficiencies.

Depending on the tools used for the diagnosis of malnutrition, up to 80% of CKD patients are malnourished [39]. Up to 60% of pre-dialysis patients are at increased risk for malnutrition, while as many as 75.9% of patients receiving HD have an increased risk [20,43,44,45,46,47]. In stark contrast, only 20% of post-kidney transplant patients are at increased risk for malnutrition [48,49]. Our results are in line with the findings of other studies, but the percentage of patients at risk of malnutrition is considerably lower than what has been reported previously [20,44,46,47,48]. This may be a result of the use of a different tool to assess nutritional status and response bias to the questionnaire. The multiple regression analysis showed that the nutritional status only influences the PCS, with no influence on the MCS or KDCS. This is consistent with the current literature. Related trials reported that the malnutrition-inflammation score (MIS) and PEW were the strongest determinants of PCS in all stages of CKD, including patients on hemodialysis [5,49]. One possible explanation for this is the fact that a dialysis diet is among the most restrictive diets, which can lead to less energy being acquired from each meal, reducing physical function [36]. After kidney transplantation, the potential nutrition shift still carries a risk of malnutrition. Lower protein intake and, consequently, a greater risk of PEW is a strong contributor to a lower PCS level and higher fatigue in kidney transplant patients, with similar findings being obtained when analyzing MIS [50,51].

### Limitations

The limitations of this study include non-random sampling and a lack of multicenter involvement, resulting in an inability to generalize results. The MNA test performed is a screening and the results predict risk rather than report incidence of malnutrition. The study was carried out at a single point in time, and does not show the dynamics of changes in the nutritional status of patients. In the future, the study should be carried out at different time points and supplemented with laboratory tests, e.g., serum pre-albumin (mg/dL), serum total iron-binding capacity (mg/dL).

## 5. Conclusions

CKD is associated with decreased QOL as well as significant morbidity and mortality, all of which are amplified in the setting of malnutrition. Moreover, malnutrition is a strong determinant of PCS score in CKD and kidney transplant patients, in which the physical function is already impacted by kidney dysfunction and treatment protocols. A systematic assessment of nutritional status and monitoring of QOL should be integrated into the standard management guidelines for CKD patients. The early detection of nutritional risk via a periodic monitoring of nutritional status, as well as the implementation of a nutritional intervention program (NIP), are essential for improving outcomes and QOL in this population. Further studies are necessary to establish standardized management guidelines that incorporate regular screening for the early detection of nutritional risk as well as an NIP designed for both the prevention and management of identified malnutrition.

## Figures and Tables

**Table 1 nutrients-14-04814-t001:** Characteristics of study groups and significance of differences between groups for selected variables.

		Group 1 N = 39		Group 2 N = 65		Group 3 N = 63		*p* ANOVA
		Mean	SD	Mean	SD	Mean	SD	
Age (years)		58.56	8.04	60.49	7.57	62.06	6.98	0.0916
Body height (m)		1.69	0.08	1.67	0.09	1.68	0.09	0.5035
Body mass (kg)		77.13	16.08	76.46	16.99	74.88	12.01	0.7323
BMI (kg/m^2^)		26.71	4.56	27.25	4.97	26.66	4.22	0.7352
Dialysis vintage (months)				109.92	91.96	33.25	35.39	-
Time since transplant (months)						117.87	92.75	-
		N	%	N	%	N	%	*p* Ch^2^
Sex	Women	20	51	27	42	28	44	0.6614
	Men	19	49	36	58	25	56	
Number of comorbidities	1–2	12	31	18	28	9	14	0.2287
	3–4	19	49	36	55	37	59	
	≥5	8	20	11	17	17	27	
Education	primary	6	16	15	23	20	32	0.1545
	secondary	13	31	27	42	24	38	
	tertiary	20	54	23	35	19	30	

**Table 2 nutrients-14-04814-t002:** Physical, mental and kidney disease component summaries from KDQoL-SF^TM^ questionnaire and nutritional assessment score in patients in each group.

		Group 1, N = 39		Group 2, N = 65		Group 3, N = 63		*p* ANOVA Kruskal Wallis	*p* POST-HOC
		Median	IQR	Median	IQR	Median	IQR		
MNA, total assessment pt		27.00	2.50	25.50	3.00	25.00	2.50	<0.0001 *	a *, b *
MNA, qualitatively N (%)	24–30	32 (82%)		48 (74%)		46 (73%)		0.5463	
	<24	7 (18%)		17 (26%)		17 (27%)			NS
KDQoL SF-total		60.70	16.90	61.90	16.40	65.10	21.35	0.1445	NS
PCS		52.50	32.50	46.25	23.75	51.88	40.00	0.0817	NS
MCS		58.21	26.71	68.54	18.79	74.29	24.42	0.0036*	b *
KDCS		57.98	16.81	66.97	19.78	59.91	16.75	0.0018*	a *, c *

MNA—Mini Nutritional Assessment, PCS—physical component summaries; MCS—mental component summaries; KDCS—kidney disease component summaries; a—group 1 vs 2; b—group 1 vs 3; c—group 2 vs 3; NS—statistically insignificant values; * *p* < 0.05.

**Table 3 nutrients-14-04814-t003:** Measurements of selected somatic parameters.

	Group 1, N = 39		Group 2, N = 65		Group 3, N = 63		
	Mean	SD	Mean	SD	Mean	SD	*p* ANOVA
Waist circumference (cm)	93.92	14.51	94.25	14.02	96.38	11.95	0.5715
Hip circumference (cm)	101.76	8.68	103.96	12.91	104.32	8.39	0.3540
WHR	0.92	0.10	0.91	0.09	0.92	0.09	0.5901

WHR—Waist–Hip Ratio.

**Table 4 nutrients-14-04814-t004:** Multivariate regression model calculated for the summary score of the questionnaire KDQoL-SF^TM^.

Dependent Variable KDQoL SF Total	Coef. b	±95% CI	*p* Value
Group	4.65	29.24–91.82	0.0002 *
Age	−0.54	1.88–7.42	0.0011 *
BMI	−0.49	−0.82–−0.26	0.0002 *
WHR	−13.59	−1.01–0.03	0.0665
MNA, total score	2.06	−38.69–11.5	0.2863
R	0.47		
Adjusted R^2^	0.19		
SE	13.25		
*p* value	<0.0001		

Group: 1, 2, 3; Age: 0—age < 65, 1—age +65; BMI—body mass index; MNA—Mini Nutritional Assessment; WHR—Waist–Hip Ratio; * *p* < 0.05.

**Table 5 nutrients-14-04814-t005:** Multivariate regression model for the PCS of the questionnaire KDQoL-SF^TM^.

Dependent Variable PCS	Coef. b	±95% CI	*p* Value
Group	−2.40	−5.59–0.80	0.1400
Age	−7.66	−12.89–−2.43	0.0044 *
BMI	−0.61	−1.18–−0.03	0.0387
WHR	34.43	6.63–62.23	0.0156 *
MNA, total score	1.58	0.60–2.55	0.0017 *
MCS	0.60	0.44–0.77	<0.0001 *
KDCS	0.22	−0.02–0.45	0.0712
R	0.73		
Adjusted R^2^	0.51		
SE	14.53		
*p* value	<0.0001		

Group: 1, 2, 3; Age: 0—age < 65, 1—age +65; BMI—body mass index; MNA—Mini Nutritional Assessment; WHR—Waist–Hip Ratio; PCS—physical component summaries; MCS—mental component summaries; KDCS—kidney disease component summaries; * *p* < 0.05.

**Table 6 nutrients-14-04814-t006:** Multivariate regression model for the MCS of the questionnaire KDQoL-SF^TM^.

Dependent Variable MCS	Coef. b	±95% CI	*p* Value
Group	5.77	3.29–8.26	<0.0001 *
Age	4.69	0.34–9.03	0.0348 *
BMI	0.20	−0.28–0.67	0.4200
WHR	−11.05	−34.25–12.14	0.3480
MNA, total score	0.32	−0.50–1.15	0.4428
PCS	0.41	0.30–0.52	<0.0001 *
KDCS	0.60	0.43–0.77	<0.0001 *
R	0.79		
Adjusted R^2^	0.60		
SE	11.93		
*p* value	<0.0001 *		

Group: 1, 2, 3; Age: 0—age < 65, 1—age +65; BMI—body mass index; MNA—Mini Nutritional Assessment; WHR—Waist–Hip Ratio; PCS—physical component summaries; MCS—mental component summaries; KDCS—kidney disease component summaries; * *p* < 0.05.

**Table 7 nutrients-14-04814-t007:** Multivariate regression model for the KDCS of the questionnaire KDQoL-SF^TM^.

Dependent Variable KDCS	Coef. b	±95% CI	*p* Value
Group	−1.96	−4.05–0.12	0.0648
Age	−4.24	−7.70–−0.79	0.0165 *
BMI	−0.04	−0.42–0.34	0.8421
WHR	−23.19	−41.40–−4.98	0.0129 *
MNA, total score	−0.25	−0.91–0.41	0.4609
PCS	0.09	−0.01–0.20	0.0712
MCS	0.38	0.27–0.49	<0.0001
R	0.71		
Adjusted R^2^	0.48		
SE	9.53		
*p* value	<0.0001		

Group: 1, 2, 3; Age: 0—age < 65, 1—age +65; BMI—body mass index; MNA—Mini Nutritional Assessment; WHR—Waist–Hip Ratio; PCS—physical component summaries; MCS—mental component summaries; KDCS—kidney disease component summaries; * *p* < 0.05.

## Data Availability

Not applicable.

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
