# Peer review of "Assessment of the Nutritional Status and Quality of Life in Chronic Kidney Disease and Kidney Transplant Patients: A Comparative Analysis"

_nutrients, 2022, doi:10.3390/nu14224814_

Round 1

Reviewer 1 Report

This is an interesting study about the nutritional status and quality of life in patients with CKD compared to patients after kidney transplantation in order to determine what factors influence nutritional status and QOL in this population. Nevertheless, the following points should be revised or corrected:

-     -  Indicate only the significant figures ( one or two) in the values of mean, standard deviantion, median and IQR in all tables and results.

-       - Do not indicate p =0.0000 in the tables. Instead of this, use p<0.0001.

-      -  Please, indicate the p-values of table 3 in table 2 with *, a, b as upper scores when the differences are significant (p<0.05; p>0.01  or p<0.001). Remove table 3.

-     -   Indicate the small sample size of each group  and that it is a unicenter study (not multicenter) as limitations of the study.

Author Response

Dear Reviewer,

We greatly appreciate your time and effort dedicated to providing feedback on our manuscript, and we are grateful for the insightful comments and valuable improvements to our paper. All the suggestions helped us to evaluate our outcomes even more precisely in order to deliver an improved, high quality scientific manuscript which we hope will now meet the high standards of Nutrients.

Comments and Suggestions for Authors

This is an interesting study about the nutritional status and quality of life in patients with CKD compared to patients after kidney transplantation in order to determine what factors influence nutritional status and QOL in this population. Nevertheless, the following points should be revised or corrected:

- Indicate only the significant figures (one or two) in the values of mean, standard deviantion, median and IQR in all tables and results.

- Do not indicate p =0.0000 in the tables. Instead of this, use p<0.0001.

- Please, indicate the p-values of table 3 in table 2 with *, a, b as upper scores when the differences are significant (p<0.05; p>0.01  or p<0.001). Remove table 3.

- Indicate the small sample size of each group  and that it is a unicenter study (not multicenter) as limitations of the study.

Answer: All suggested revisions have been made.

Thank you for your feedback. The manuscript has been revised and all of your suggestions have been incorporated.

Once again, thank you very much for your review.

Best regards,

Authors

Reviewer 2 Report

If no statistical difference between groups, then difficult to state the nutritional status plays a major role in patient outcomes. At the same time, it is evident that malnutrition has a major physical and psychological impact on QoL based on my 50 years of observation in the CKD population. It would be worthwhile to assess change in annualized BMI and estimated protein intake (EPI) in these 3 cohorts (for a 3 year period). Then compare overall annualized hospitalization rates and mortality rates in these 3 groups to determine if BMI and EPI are independent variables on outcomes. This would give a snapshot of the critical impact of nutrition on morbidity and mortality.

Author Response

Dear Reviewer,

We greatly appreciate your time and effort dedicated to providing feedback on our manuscript, and we are grateful for the insightful comments and valuable improvements to our paper. All the suggestions helped us to evaluate our outcomes even more precisely in order to deliver an improved, high quality scientific manuscript which we hope will now meet the high standards of Nutrients.

Comments and Suggestions for Authors

If no statistical difference between groups, then difficult to state the nutritional status plays a major role in patient outcomes. At the same time, it is evident that malnutrition has a major physical and psychological impact on QoL based on my 50 years of observation in the CKD population. It would be worthwhile to assess change in annualized BMI and estimated protein intake (EPI) in these 3 cohorts (for a 3 year period). Then compare overall annualized hospitalization rates and mortality rates in these 3 groups to determine if BMI and EPI are independent variables on outcomes. This would give a snapshot of the critical impact of nutrition on morbidity and mortality.

Thank you for your comments and suggestions. We agree, that investigating the effects of change in annualized BMI and EPI on outcome would further elucidate the impact of nutrition on morbidity and mortality. We are expanding on our findings in this manuscript with further data collection and will include these variables in our next manuscript.

We appreciate your review of this manuscript and your insight for further analysis.

Best regards,

Authors